# Alternative Pathways to Ciprofloxacin Resistance in *Neisseria* *gonorrhoeae*: An In Vitro Study of the WHO-P and WHO-F Reference Strains

**DOI:** 10.3390/antibiotics11040499

**Published:** 2022-04-08

**Authors:** Natalia González, Saïd Abdellati, Irith De Baetselier, Jolein Gyonne Elise Laumen, Christophe Van Dijck, Tessa de Block, Chris Kenyon, Sheeba Santhini Manoharan-Basil

**Affiliations:** 1STI Unit, Department of Clinical Sciences, Institute of Tropical Medicine, 2000 Antwerp, Belgium; jlaumen@itg.be (J.G.E.L.); cvandijck@itg.be (C.V.D.); ckeyon@itg.be (C.K.); 2Clinical Reference Laboratory, Department of Clinical Sciences, Institute of Tropical Medicine, 2000 Antwerp, Belgium; sabdellati@itg.be (S.A.); idebaetselier@itg.be (I.D.B.); tdeblock@itg.be (T.d.B.); 3Laboratory of Medical Microbiology, University of Antwerp, 2610 Wilrijk, Belgium; 4Division of Infectious Diseases and HIV Medicine, University of Cape Town, Anzio Road, Observatory, Cape Town 7700, South Africa

**Keywords:** *N. gonorrhoeae*, fluoroquinolone, AMR, resistance, ciprofloxacin

## Abstract

Emerging resistance to ceftriaxone and azithromycin has led to renewed interest in using ciprofloxacin to treat *Neisseria gonorrhoeae*. This could lead to the rapid emergence and spread of ciprofloxacin resistance. Previous studies investigating the emergence of fluoroquinolone resistance have been limited to a single strain of *N. gonorrhoeae*. It is unknown if different genetic backgrounds affect the evolution of fluoroquinolone resistance in *N. gonorrhoeae*, as has been shown in other bacterial species. This study evaluated the molecular pathways leading to ciprofloxacin resistance in two reference strains of *N.*
*gonorrhoeae*—WHO-F and WHO-P. Three clones of each of the two strains of *N.*
*gonorrhoeae* were evolved in the presence of ciprofloxacin, and isolates from different time points were whole-genome sequenced. We found evidence of strain-specific differences in the emergence of ciprofloxacin resistance. Two out of three clones from WHO-P followed the canonical pathway to resistance proceeding via substitutions in GyrA-S91F, GyrA-D95N and ParC. None of the three WHO-F clones followed this pathway. In addition, mutations in *gyrB*, *uvrA* and *rne* frequently occurred in WHO-F clones, whereas mutations in *yhgF*, *porB* and *potA* occurred in WHO-P.

## 1. Introduction

The World Health Organization (WHO) has categorized *Neisseria gonorrhoeae* as being at high risk for the emergence of further antimicrobial resistance (AMR) [1]. In particular, the emergence of combined gonococcal resistance to ceftriaxone and azithromycin has renewed interest in reintroducing fluoroquinolones (FQs) as therapeutic agents [1,2]. The resistance mechanisms developed by *N. gonorrhoeae* to ciprofloxacin (CIP) are due to point mutations in the quinolone resistance determining region (QRDR) of the *gyrA* and *parC* genes [3,4,5,6]. One study found that CIP could be successfully used to treat gonococcal isolates that were negative for the S91F mutation in DNA gyrase subunit A [2]. An understandable concern of this approach is that this will result in a renewed increase in fluoroquinolone resistance in *N. gonorrhoeae*.

The initial introduction of fluoroquinolones as a treatment for gonorrhoea was followed by an explosive increase in the prevalence of resistance to FQs [1,7]. The prevalence of resistance has declined in recent years in a number of locales, and it would be prudent only to reintroduce FQ to treat gonorrhoea in a way that minimizes the selection pressure for resistance [8]. An important unknown in this regard is if all gonococcal strains have the same propensity to develop FQ resistance. Studies in *Mycobacterium tuberculosis*, *Staphylococcus aureus* and *Clostridium difficile* have found large variations in this propensity [9,10,11]. For example, in the case of *M. tuberculosis*, in vitro experiments have confirmed that differences in the genetic background can result in two orders of magnitude differences in the frequency of the emergence of FQ resistance [9].

In a recent phylogenetic analysis of a global collection containing 17,871 isolates of *N. gonorrhoeae*, we found that the prevalence of FQ resistance varied considerably according to multilocus sequence type (MLST) [12]. About twenty-five MLSTs were represented by 100 or more isolates. If the GyrA-S91F substitution is used to define ciprofloxacin resistance, then of these 25 MLSTs, 13 MLSTs were predominantly resistant (median 97.4% [IQR 95.6–98.4%] resistance). The remaining 12 MLSTs were predominantly susceptible to ciprofloxacin (median 2.7% [IQR 0.2–10.1%] resistance). The same was true for high-level resistance (HLR; MIC to ciprofloxacin ≥16 mg/L). Of the 2384 ciprofloxacin HLR isolates detected, half of the isolates (50.4%) belonged to ST1901, followed by ST7363 (11.49%) and ST1579 (2.5%).

The molecular pathway to ciprofloxacin resistance in *N. gonorrhoeae* has been previously been assessed in vitro [3]. However, this study was limited to a single strain (FA-19), was not performed in duplicate, and the molecular sequencing was limited to the *gyrA/B* and *parC/E* genes [3]. In preliminary work, we noted that despite identical ciprofloxacin MICs (0.004 mg/L), the WHO-P strain of *N. gonorrhoeae* appeared to acquire resistance to ciprofloxacin more rapidly than WHO-F. Therefore, in this work, we aimed to assess if there is a difference between WHO-F and WHO-P in (1) order of acquisition of mutations in the target genes, (2) time to ciprofloxacin resistance, and (3) the molecular pathway to ciprofloxacin resistance.

## 2. Methods

### 2.1. WHO-Reference Strains, Genetic Characteristics and Comparative Genomics

The strains used in this study were *N. gonorrhoeae* WHO reference strains (WHO-F and WHO-P), both susceptible to ciprofloxacin with a minimum inhibitory concentration (MIC) of 0.004 mg/L. The genetic background of the two strains are as follows: (i) WHO-F belongs to *porB1a* serogroup with a WT *mtrR* promoter region, *rpoB* and *rpsJ* genes (ii) WHO-P belongs to *porB1b* serogroup, with A to C substitution in *mtrR* promoter region, and substitutions in RpoB- H552N and RpsJ-V57M [13]. Furthermore, the clades of WHO-F and WHO-P were further investigated to delineate the molecular pathways to FQ resistance using the previously defined whole genome MLST (wgMLST) and core genome MLST (cgMLST) schemes according to Manoharan-Basil et al. 2022. Briefly, WGS data that comprised of 17, 871 *N. gonorrhoeae* isolates were analyzed using chewBBACA version 2.8.5 [14]. A training file using the complete genome of *N. gonorrhoeae* FA1090 was created using Prodigal [15] and was followed by creating a study specific schema using two complete *N. gonorrhoeae* genomes (FA1090 and MS11). A FASTA file for each coding sequence (CDS) was generated, followed by the creation of wgMLST loci. The cgMLST loci were then extracted from the wgMLST loci and visualized using a grape tree [16].

### 2.2. Plating Experiment and MIC Determination

All of the strains were grown on a gonococcal (GC) agar plate (Gonococcal Medium Base, BD Difco™) supplemented with 1% IsoVitaleX (BD BBL™) and incubated at 36 °C in an atmosphere of 5% CO_2_. Three independent clonal lineages of WHO-F (henceforth referred to as WHO-F_1_, WHO-F_2_ and WHO-F_3_) and WHO-P (henceforth referred to as WHO-P_1_, WHO-P_2_ and WHO-P_3_) strains were evolved from single colonies grown on a GC agar plate. A single colony from each strain was inoculated on a GC agar plate containing 0.004 mg/L ciprofloxacin and incubated at 36 °C with 5% CO_2_. After visible growth was attained, typically after 36–72 h, one colony was inoculated onto a GC agar plate in which the drug concentration had been increased two-fold relative to the previous concentration. The selection was continued until a ciprofloxacin (CIP) MIC concentration of 32 mg/L was attained, or no visible growth was seen. The cultures from each time point were stored in skimmed milk supplemented with 20% of glycerol and stored at −80 °C. The number of passages for each strain is presented in Table 1.

MICs were determined using E-test strips (BioMerieux, Marcy-l’Étoile, France) on GC agar according to the manufacturer’s instructions. The MIC for CIP ≥ 16 mg/L, >0.06 mg/L, 0.03 ≤ 0.06 and <0.03 mg/L were classified as being high-level resistant (HLR), resistant (R), intermediate (I) and susceptible (S) to CIP, respectively [12,17]. Geometric mean (GM) MIC are provided where appropriate. The doubling time was estimated to be ~60 min [18].

### 2.3. Whole-Genome Sequencing (WGS) and SNP Analysis

In this study, 11 clones from WHO-F [WHO-F_1_ (n = 3); WHO-F_2_ (n = 4); WHO-F_3_ (n = 4)] and 12 clones from WHO-P [WHO-P_1_ (n = 4); WHO-P_2_ (n = 3); WHO-P_3_ (n = 5)] strains were subjected to WGS (Table 1). Genomic DNA was extracted using the MasterPure Complete DNA and RNA Purification Kit (Epicenter, Madison, WI, USA) and eluted in nuclease-free water. DNA was outsourced for WGS (GENEWIZ, Germany and Eurofins, Leipzig, Germany) and was sequenced on an Illumina instrument using 150 bp paired-end sequencing chemistry (Illumina, San Diego, CA, USA). The quality of the raw reads was assessed using FASTQC [19]. The raw reads were trimmed for quality (Phred ≥ 20) and length (≥32 bases) using trimmomatic (v0.39) [20]). The processed reads from the baseline isolates were assembled using Shovill (v1.0.4) [21], which uses SPAdes for the *denovo* assembly (v3.14.0). The parameters used for *denovo* assembly are as follows: —trim —depth 150 —opts —isolate. The quality of the *denovo* contigs was evaluated using Quast (v5.0.2) [22]. Finally, the draft genome was annotated using Prokka (v1.14.6) [23]. The quality controlled reads were mapped to baseline draft genome using BWA MEM and single nucleotide polymorphisms (SNPs) were determined using freebayes implemented in Snippy (v4.6.0) with default parameters (10× minimum read coverage and 90% read concordance at the variant locus) [24,25,26]. The raw reads generated are deposited at https://www.ncbi.nlm.nih.gov/sra/PRJNA815938 (accessed on 14 March 2022).

Genetic characterization of the relevant genes associated with ciprofloxacin resistance in WHO-F, WHO-P, and global *Neisseria* spp. collection.

The putative SNPs identified in the relevant genes associated with ciprofloxacin resistance were further examined in the global collection comprising the genomes of *N. gonorrhoeae* (n = 17,871), including the WHO-F and WHO-P, and commensal *Neisseria* spp. (n = 1136) whose provenance and metadata are described elsewhere [12]. 

### 2.4. Statistical Analysis

Stata 16.0 (StataCorp, College Station, TX, USA) was used for all analyses. A *p*-value of <0.05 was considered statistically significant. Mann-Whitney *U*-test was conducted to test for significant changes in the MIC.

## 3. Results

### 3.1. Association of WHO-F and WHO-P and Ciprofloxacin MICs

WHO-F belongs to ST10934, whereas WHO-P belongs to ST8127 (Figure 1). ST10934 (n = 9, including WHO-F) originated from ST7359 (n = 653) and ST8127 (n = 6, including WHO-P) originated from ST1580 (n = 560). Isolates belonging to ST7359 were all *gyrA* and *parC* wild type (WT). Out of 560 isolates belonging to ST1580, 23 isolates had mutations conferring resistance to ciprofloxacin, 257 were WT, and 280 isolates had no available data. Out of these 22 isolates, different combinations of mutations in the target genes were identified and are as follows: twelve isolates had substitutions in GyrA-S91F, GyrA-D95A and ParC-S87R (GM MIC—10.5 mg/L), four isolates had the GyrA-S91F and ParC-D86N substitutions (GM MIC—0.38 mg/L), two isolates had the GyrA-S91F, GyrA-D95G, ParC-E91Q substitutions (GM MIC—0.07 mg/L), two isolates had the GyrA-S91F, GyrA-D95G and ParC-S87R (GM MIC—4 mg/L), one isolate had the substitutions in GyrA-S91F, and GyrA-D95G (MIC—8 mg/L) and one isolate had only the ParC-S87R substitutions (MIC—0.02 mg/L).

### 3.2. In-Vitro Selection of Ciprofloxacin Resistance in WHO-F and WHO-P

In all six experiments, clones of the WHO-F and WHO-P strains acquired ciprofloxacin resistance (MIC > 0.06 mg/L). Only one clone, WHO-P_3,_ acquired high-level ciprofloxacin resistance and attained a MIC of 32 mg/L by day 18 (~432 generations), representing an increase in MIC compared to baseline of about 8000-fold (Table 1). 

### 3.3. Mutations in Fluoroquinolone Target Genes (gyrA, gyrB, parC and parE)—GyrA S91F Pathway Is Associated with Higher Ciprofloxacin MICs 

Mutations were detected in 9 out of the 11 clones of the WHO-F strain [WHO-F_1_ (2/3); WHO-F_2_ (3/4); WHO-F3 (4/4)] and in 9 out of the 12 clones of the WHO-P strains [WHO-P_1_ (3/4); WHO-P_2_ (2/3); WHO-P_3_ (4/5)].

The genetic mechanisms underlying the antibiotic resistance in the evolved CIP resistant isolates are as follows:
I.WHO-F

WHO-F_1_ and WHO-F_3_ developed resistance after five passages, whereas the isolates of WHO-F_2_ attained resistance after four passages (Table 1).
(a)WHO-F_1_ & WHO-F_3_: WHO-F_1_ and WHO-F_3_ acquired the GyrA-D95N substitution at days 5 (~120 generations) and 6 (~144 generations), respectively (MIC−0.032 mg/L). The highest MICs attained by these clones was 0.125 mg/L.(b)WHO-F_2_: WHO-F_2_ acquired the S91Y substitution in the quinolone resistance-determining regions (QRDR) in GyrA at day 2 (~48 generations, MIC−0.064 mg/L) which was followed by additional substitutions in GyrA-D95N (MIC−0.38 mg/L; Day-13 and MIC−12 mg/L; Day-18) and ParC- E91K (MIC−12 mg/L; Day 18).

Additional substitutions outside the QRDR region were also observed and are as follows: GyrA-D80Y in WHO-F_1_ (MIC−0.125 mg/L; Day 12) and ParC-R537S in WHO-F_3_ (MIC-0.125 mg/L; Day 11).

II.WHO-P

WHO-P_1_ took three passages to reach a MIC of 0.125 mg/L while isolates from WHO-P_2_ and WHO-P_3_ required one passage of ciprofloxacin exposure to attain the same MIC (Table 1).
(a)WHO-P_1_: An insertion in GyrB (S467_G468ins) emerged by day 2 in WHO-P_1_ (MIC−0.008 mg/L) and persisted till day 3 (MIC−0.012 mg/L). On day 6, WHO-P_1_ acquired the D95N substitution in GyrA, and its MIC increased to 0.125 mg/L, which was the highest MIC attained by WHO-P_1_.(b)WHO-P_2_ & WHO-P_3_: The WHO-P_2_ and WHO-P_3_ clones acquired the GyrA-S91F substitution by day 2 (MIC−0.125 mg/L), followed by substitutions in ParC (ParC-D86N substitution in WHO-P_2_ by day 16 [MIC 4 mg/L] and ParC-R537L substitution outside the QRDR at day 11 [MIC−0.125 mg/L]). WHO-P_2_ and WHO-P_3_ attained higher MICs (4 mg/L and 32 mg/L, respectively) than WHO-P_1_ (0.125 mg/L).

Among all WHO-F and WHO-P clones, those that acquired the S91F/Y substitution in GyrA (WHO-F_2_, WHO-P_2_ & WHO-P_3_) attained higher ciprofloxacin MICs (12, 4 & 32 mg/L, respectively) than the clones that did not acquire this mutation (WHO-F_1_, WHO-F_3_ & WHO-P_1_, all MIC 0.125 mg/L; *p* = 0.037). The S91F/Y clones also survived for longer (16, 18 & 21 days) than those that did not acquire S91F/Y (7, 12 & 13 days; Table 1; *p* = 0.049).

### 3.4. Mutations in gyrB, uvrA and rne Frequently Occurred in WHO-F Strains, Whereas Mutations in yhgF, porB and potA Occurred in WHO-P during the Selection for Ciprofloxacin Resistance 

WGS analysis revealed that in addition to acquiring the known resistance associated mutations (RAMs- GyrA-S91Y/F, GyrA-D95N, ParC-E91K and ParC-D86N) additional substitutions were differentially detected in WHO-F and WHO-P. In WHO-F, substitutions in *gyrB* [WHO-F_1_ (n = 1), WHO-F_3_ (n = 3)], *rne* [WHO-F_1_ (n = 2)] and *uvrA* [WHO-F_1_ (n = 1), WHO-F_3_ (n = 2)] were identified (Figure 2 A, C, E), whereas in WHO-P substitutions were detected in *yhgF* [WHO-P_2_ (n = 2), WHO-P_3_ (n = 2)], *porB* [WHO-P_1_ (n = 1), WHO-P_3_ (n = 1)] and *potA* [WHO-P_3_ (n = 1)] (Figure 2B,D,F).

#### 3.4.1. Mutations in *gyrB*, *uvrA* and *rne* in WHO-F

Two clones (WHO-F_1_ and -F_3_) acquired substitutions in UvrA. In both cases, this involved a frameshift (fs) caused by a deletion (del) (987–997del GAC TTC AAT CGC; Figure 2A,E), UvrA-M329fsdel, leading to a truncated protein. These substitutions occurred at the same time as P739H substitutions in GyrB and D95N substitutions in GyrA that were associated with intermediate resistance (MICs—0.032 to 0.125 mg/L). Neither WHO-F_1_ nor WHO-F_3_ acquired the S91F GyrA substitution. The P739H substitution in GyrB was first observed in WHO-F_3_ (n = 3) on day one and in WHO-F_1_ (n = 1) on day five.

In two of the four-time points at which the P739H substitution in GyrB was detected in WHO-F_3_, no other substitutions were found (Figure 2E). These time points were the first two time points for WHO-F_3_. The ciprofloxacin MICs for these time points were not elevated compared to baseline (0.004 mg/L on baseline and day 1 and 0.006 mg/L on day 2). 

In contrast, WHO-F_2_ did not acquire these substitutions in GyrB or UvrA but did acquire a frameshift mutation in Rne (M329fs) at two-time points that were contemporaneous with the S91Y substitution in GyrA (Figure 2C).

#### 3.4.2. Mutations in *lldD*, *porB*, *potA* and *yhgF* in WHO-P

In two WHO-P_1_ clones, an insertion in GyrB (S467_G468ins) was observed at the same time a missense mutation in LldD-S114Y that emerged at day 2 (MIC–0.008 mg/L) and persisted until the MIC had increased three-fold (MIC–0.012 mg/L; Figure 2B). E127K substitutions were found to occur in PorB in WHO-P_1_ and WHO-P_3_ clones coincident with D95N substitutions in GyrA (MIC-0.125 to 32 mg/L; Figure 2B,F). The YhgF-A414V substitution was observed in two and four isolates belonging to WHO-P_2_ and WHO-P_3_ lineages, respectively (Figure 2D,F). This YhgF substitution was always accompanied by the GyrA-S91F substitution (MIC baseline-0.004 mg/L, mutants–0.125 mg/L, 0.5 mg/L, 4 mg/L and 32 mg/L). 

The PotA-G280D substitution was found in the last two clones belonging to the WHO-P_3_ lineage and was accompanied by a number of known RAMs such as S91F and D95N in GyrA as well ParC-R537L and ParE- P456S substitutions (MIC—32 mg/L; Figure 2F). 

### 3.5. Other Mutations

A number of other mutations at single time points were identified and are as follows: (i) a duplication (dup) in transcriptional regulatory protein, ZraR-D5fs (12_13dupCG) that was detected in sample WHOP_2_ (MIC-4 mg/L) after 16 days of experiment; (ii) a deletion in pilin glycosyltransferase, PglA-R253fs (756delG) was identified after a day of exposure in WHO-F_1_ (MIC–0.004 mg/L); and (iii) a deletion in ribosomal protein L11 methyltransferase, PrmA-E93del (276–277 del GC) in WHOF_1_ at day 12 (MIC–0.125 mg/L). The relevance of these mutations to the genesis of ciprofloxacin resistance requires further experimentation. 

### 3.6. Genetic Characterization of Relevant Genes (potA, rne, uvrA and yhgf) in the Genomes of WHO-F and WHO-P

The baselines genomes of WHO-F and WHO-P were aligned, and the following amino acid changes were observed in the relevant genes considered in this manuscript. (i) PotA: Amino acid at positions 34 and 75 in WHO-P were N (Asn) and D (Asp), whereas, in WHO-F, they were D (Asp) and N (Asn), respectively. (ii) RnE: In WHO-P amino acids at positions 399 and 437 were A (Ala) and (R) Arg (R), whereas, in WHO-F, they were V (Val) and S (Ser) (iIi) UvrA: In WHO-P the amino acids at positions 163, 619, 931, and 935 were (G) Gly, D (Asp), (E) Glu and (V) Val, whereas in WHO-F they were A (Ala), (G) Gly, Q (Gln) and (I) Ile, respectively (iv) YhgF: Amino acids at positions 96 and 116 in WHO-P were A (Ala) and R (Arg), and in WHO-F they were T (Thr) and H (His), respectively (Appendix A).

### 3.7. gyrB, parC, porB and yhgF Mutations in the GLOBAL Collection

The GyrB-P739H substitution was not observed in the 17,871 *N. gonorrhoeae* isolates. This substitution was, however, identified in two commensal *Neisseria* spp.—*N. brasiliensis* (PATRIC ID-2666100.4) and *Neisseria* sp. (PubMLST ID-94179). The ParC-R537S/L substitutions were not observed in any of the *Neisseria* spp. (n = 19,007). ParC-R53C and ParC-R537H substitutions were observed in three (GM MIC—0.75 mg/L) *N. gonorrhoeae* isolates. The PorB-E127K substitution in loop 3 was not present in any of the *Neisseria* spp. However, the PorB-E127D substitution was identified in three *N. gonorrhoeae* (MIC– not available) and 102 commensal *Neisseria* spp. (MIC- not available) Furthermore, PorB-E127G (n = 1, MIC–32 mg/L) substitution was found in one *N. gonorrhoeae* isolate (Pathogenwatch ID—ECDC-FR13-066). 

## 4. Discussion

The impacts of different genetic backgrounds in *N. gonorrhoeae* on the evolution of FQ resistance are poorly understood. In this study, we evaluated the pathway to the FQ resistance of two gonococcal WHO reference strains with the same ciprofloxacin MIC (0.004 mg/L). Our global phylogenetic analysis revealed that none of the 653 isolates from ST7359 from which WHO-F emerged had acquired the FQ RAMs. In contrast, 21 out of 560 isolates from the WHO-P lineage associated ST1580 had acquired FQ RAMS. 

Our experimental findings are compatible with strain- and pathway-specific differences in the emergence of ciprofloxacin resistance. 

### 4.1. Pathway Specific Differences

Two clones of WHO-P followed the GyrA S91F pathway and one clone of WHO-F followed the similar GyrA S91Y pathway. These three clones attained higher ciprofloxacin MICs than three other clones from WHO-P and -F that acquired D95N- and not S91F- substitutions. 

Furthermore, two of the three clones that followed the S91F/Y pathway went on to acquire known RAMs in ParC—D86N in WHO-P_2_ and E91K in WHO-F_2_—compared to none of the three clones that did not acquire S91F/Y substitutions. These findings are compatible with previous studies that have found that the development of FQ resistance in *N. gonorrhoeae* commences with the GyrA S91F substitution and that subsequent increases in MIC are related to D95N substitutions in GyrA and various mutations in ParC [3,27]. We found two other substitutions in ParC that were outside the QRDR but may be associated with increased MICs—R537L in WHO-P_3_ and R537S in WHO-F_3_. These substitutions were not found in our global dataset, and the relevance of these substitutions thus requires further investigation.

### 4.2. Strain-Specific Differences 

Whilst one clone of WHO-F followed the S91Y GyrA pathway, this substitution was found far less commonly than the S91F substitution in an analysis of 17, 871 isolates of *N. gonorrhoeae* from around the world (n = 27 versus n = 17,871, respectively). This, plus the finding noted above that the GyrA S91F substitution has not been detected in the WHO-F-like sequence types, suggests that the probability of the S91F substitution emerging in WHO-F may be lower than in other strains such as WHO-P. In addition, we found strain-specific differences in the mutations outside the topoisomerase genes. In WHO-F, mutations were detected in *gyrB*, *uvrA,* and *rne*, whereas in WHO-P, mutations were detected in *yhgF*, *porB,* and *potA* (Figure 3).

### 4.3. gyrB, uvrA and rne Mutations Detected in WHO-F

In WHO-F_1_ and -F_3_, the D95N substitution in GyrA was accompanied by a P739H substitution in GyrB and a M329 frameshift deletion in UvrA. This substitution in GyrB is outside the QRDR and not observed in the 17,871 isolates, and it may therefore reflect a transient mutation, appearing only temporarily and being lost at later stage due to fitness costs or other factors. 

The nucleotide excision repair (NER) gene, *uvrA*, is part of the SOS regulon in many bacteria that catalyzes the recognition and processing of DNA damage [28,29,30,31]. Ciprofloxacin has been shown to increase the transcription of *uvrA* via the SOS pathway which could play a role in facilitating the subsequent acquisition of antimicrobial resistance [29]. Further experimentation is necessary to better characterize the role of the frameshift mutation in *uvrA* in the genesis of ciprofloxacin resistance in *N. gonorrhoeae*. Notably, in the baseline isolates the amino acids at positions 163, 619, 931 and 935 varied between WHO-F (A163, G619, Q931 and I935) and WHO-P (G163, D619, E31 and V935).

By way of contrast, the pathway to ciprofloxacin resistance in WHO-F_2_ included a frameshift mutation in *rne* (Rne-D661fs), ribonuclease E, resulting in a premature stop codon (Figure 2C). In *E. coli* RNase deletion or inactivation precludes the normal initiation of the SOS response [32]. 

### 4.4. yhgF, porB and potA Mutations Detected in WHO-P

The A414V substitution in YhgF was detected at the first time point when ciprofloxacin MICs increased in both WHO-P_2_ and -P_3_ and persisted until the end of both experiments. YhgF has been shown to play a role in *E. coli’s* ability to survive ionizing radiation [33]. A genetic interaction screen also established that YhgF has interactions with a number of proteins involved in translation and ribosome biogenesis, such as S1 [34]. For example, Δ*yhgF* mutants exhibit increased levels of stop codon readthrough [34]. In our experiments, there was 100% concordance between the detection of this A414V substitution in YhgF and the S91F substitution in GyrA. This contrasts with the complete absence of the A414V substitution in YhgF in the global collection of *N. gonorrhoeae*, a high proportion of whom have the S91F substitution in GyrA. One way to explain the apparent temporary emergence of the A414V substitution is that it acts as a stepping stone to FQ resistance. Gomez et al. have produced compelling experimental evidence that mutations in ribosomal proteins act as stepping stones to FQ resistance in *Mycobacterium smegmatis* [35]. They found that ciprofloxacin exposure first selected for mutations in ribosomal proteins, which facilitated the subsequent acquisition of resistance-associated mutations in *gyrA*. The ribosomal mutations conferred a fitness cost and thus were lost at some point after acquiring the *gyrA* mutations. Further experiments are required in *N. gonorrhoeae* to test this hypothesis. 

The PorB-E127K substitution was observed at the last time point of WHO-P_1_ and the last two time points of WHO-P_3_. In both cases, it emerged at the same timepoint as the D95N substitution in GyrA. Once again, it was not found in any of the global collections. The *porB* gene encodes an outer membrane porin that has two mutually exclusive alleles—*PorB1a* in WHO-F and *PorB1b* in WHO-P [36]. Isolates with PorB1a tend to be more susceptible to penicillin and tetracycline [37]. Loop 3 of *PorB1b* folds into the barrel of the porin, constricting the pore. Amino acid substitutions in this loop, such as G120K and A121D, result in reduced susceptibility to cephalosporins, penicillin and tetracyclines [37]. Although speculative, it is possible that the E127K substitution, which is also in loop 3, has a similar effect for FQ. The fact that this substitution was not detected in the global collections would once again be compatible with this substitution having a fitness cost.

Polyamines are involved in several cellular processes such as energy metabolism, oxidative stress tolerance, biofilm formation, and iron transport [38,39,40]. PotA is a cytoplasmic protein with an ATP-binding motif that couples ATP hydrolysis to translocation of polyamines, part of the *potABCD* operon, encoding the spermidine-preferential uptake system PotABCD [41]. Mutation in *potA* (PotA-Q208L) alters the ATPase activity of the transporter, generating a 4-fold increase in gentamicin in *E.*
*coli* [42]. In our experiments, we found that substitution in PotA-G280D was detected at two-time points when WHO-P_3_ had acquired HLR (MIC-32 mg/L).

There are a number of limitations to this study. We tested the pathways to FQ resistance in a limited number of sequence types. We did not conduct knockout/complementation studies to assess the biological effect of the various mutations detected. Neither were fitness costs or cross-resistance to other antimicrobials assessed. These limitations notwithstanding, we found strain and pathway-specific variations in the genesis of FQ resistance in *N. gonorrhoeae*. WHO-F type strains may be less prone to develop ciprofloxacin resistance. This finding suggests that the proposed reintroduction of ciprofloxacin for the treatment of gonorrhea may result in less FQ resistance in regions where WHO-F type strains are more prevalent [12]. Hence, surveillance of *N. gonorrhoeae* genotypes may play an ancillary role in the safe reintroduction of FQ for the treatment of gonorrhea.

## Figures and Tables

**Figure 1 antibiotics-11-00499-f001:**
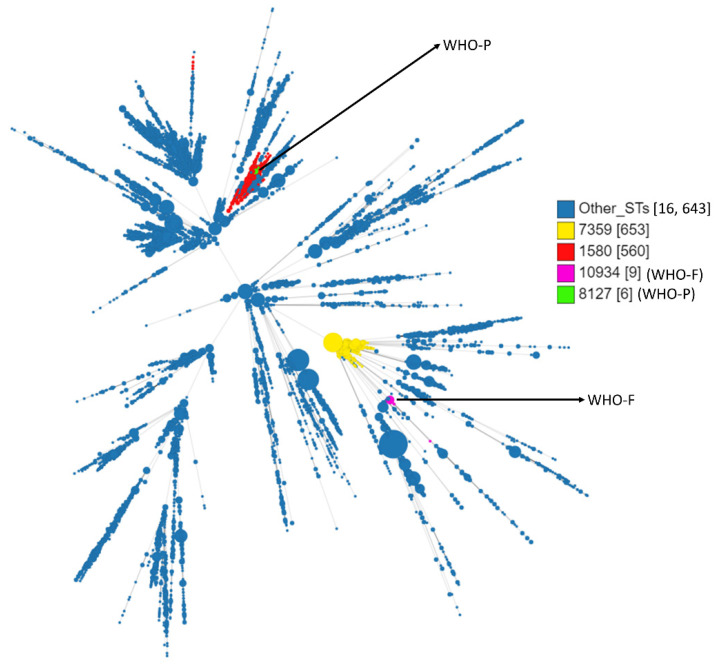
Minimum spanning tree comparing core-genome allelic profiles with MLST resulting in isolates with similar allelic profiles forming clusters. Isolates are displayed as circles. The size of each circle indicates the number of isolates of this particular type. Numbers in brackets refer to the number of isolates. WHO-F (ST10934) is denoted in magenta colour and WHO-P (ST8127) in green color.

**Figure 2 antibiotics-11-00499-f002:**
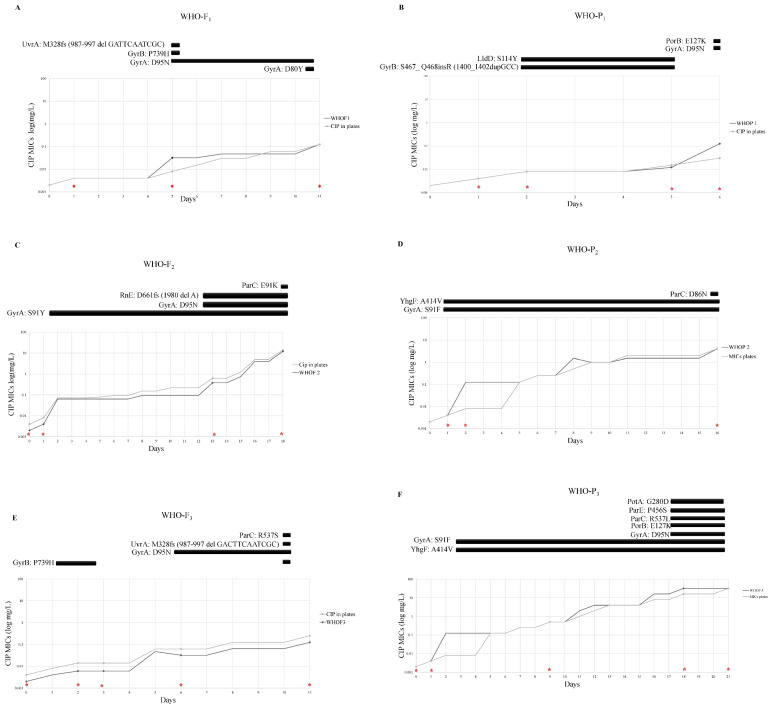
Ciprofloxacin resistance evolution of three *N. gonorrhoeae* clones of WHO-F (**A**,**C**,**E**) and three clones of WHO-P (**B**,**D**,**F**) over time. The MIC’s of ciprofloxacin were tested using E-test (bioMerieux, Marcy-l’Étoile, France) once there was visible growth. Once there was visible growth, the ciprofloxacin MIC’s were tested using E-Test (bioMerieux, Marcy-l’Étoile, France) and the next plate was inoculated. The initial MIC of all clones was 0.004 mg/L. The red cross indicates the sampling time that was subjected to WGS. The black squares represent the duration of the mutation noted next to them for as long as the days displayed in the axis below them.

**Figure 3 antibiotics-11-00499-f003:**
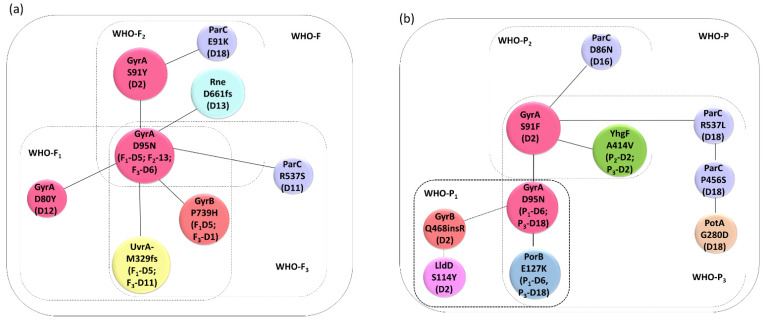
Association network of genes putatively associated with ciprofloxacin resistance in (**a**) WHO-F and (**b**) WHO-P strains. The clones and the days are denoted in parenthesis.

**Table 1 antibiotics-11-00499-t001:** Evolution of ciprofloxacin resistance over time. The isolates that were subjected to WGS are colored in grey. Asterisk denotes bacterial cell death. X: data not recorded. Asterisk denotes bacterial cell death.

WHO Reference Strains	CIP MIC (mg/L)																					Total No. of Passages
WHO-F1	0.004	X	X	X	X	0.032	0.032	0.047	X	0.047	X	0.125	*									5
WHO-F2	0.004	0.064	X	X	0.064	0.064	X	0.094	X	0.094	X	X	0.38	X	0.75	4	X	12	*			9
WHO-F3	0.004	0.006	X	X	0.047	0.032	X	0.064	X	X	0.125	*										5
WHO-P1	0.004	0.008	X	X	0.012	0.125	*															3
WHO-P2	0.004	0.125	X	X	0.125	0.25	X	1.5	1	X	1.5	X	X	X	X	4	*					7
WHO-P3	0.004	0.125	X	X	0.125	X	0.5	X	X	X	2	4	4	X	X	16	X	32	X	X	32	9
Days	1	2	3	4	5	6	7	8	9	10	11	12	13	14	15	16	17	18	19	20	21	

## Data Availability

WGS sequences: https://www.ncbi.nlm.nih.gov/sra/PRJNA815938 (accessed on 14 March 2022).

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
