# Peer review of "Alternative Pathways to Ciprofloxacin Resistance in Neisseria gonorrhoeae: An In Vitro Study of the WHO-P and WHO-F Reference Strains"

_antibiotics, 2022, doi:10.3390/antibiotics11040499_

Round 1

Reviewer 1 Report

I have read this manuscript with great interest, since the data on Neisseria gonorrhoeae resistance to antimicrobial drugs are very important both for therapy and surveillance. The reuse of ciprofloxacin for treatment of gonococcal infections verified as ciprofloxacin susceptible offers several benefits, including an effective oral treatment for both urogenital and
extragenital infections, limited side effects, a reduced selection pressure for resistance to first-line ceftriaxone and azithromycin. The study by Gonzalez et al evaluated the molecular pathways leading to ciprofloxacin resistance in reference WHO strains. Authors assessed genetic backgrounds in N. gonorrhoeae strains after their cultivation on ciprofloxacin-contained medium. In addition to the long-described mechanisms of the N. gonorrhoeae resistance aquisition to fluoroquinolones in gyrA and parC genes, the authors discovered new candidate pathways involved in the formation of resistance. Moreover, the authors described strain-specific differences in these pathways. The obtained results are original and published for the first time. Hope, that the following questions can improve the manuscript.

  1. Introduction. Please clarify the choice of WHO reference strains - why these two strains  (WHO-P and WHO-F) were taken.
  2. Line 52. It is not clear what sequence types are mentioned: NG-MAST, MLST, cgMLST ? Could the authors clarify and write it explicitly.
  3. The error distribution profile in whole genome sequencing is not random. SNPs identifyed by WGS are usually confirmed by Sanger sequencing so that the reader can determine the validity of each nucleotide polymorphism found. Did authors performs Sanger sequencing ?
    There are also no data on the statistical significance of the detected SNP (coverage at the site of a significant SNP/P-value/Q score).
  4. The occurrence of mutations is a random process, therefore it is not clear why a statistical assessment of the significance of the difference in mutagenesis pathways on such a small sample was not performed.
  5. Authors may clarify how many clones of each strain contained detected mutations (in total, 11 WHO-F clones and 12 WHO-P clones were subjected for WGS, lines 100-101).
  6. Line 94. Please remove ';' after CIP.
  7. Line 109. Please, add the reference for SPAdes: Bankevich A, Nurk S, Antipov D, Gurevich AA, Dvorkin M, Kulikov AS, et al. SPAdes: a new genome assembly algorithm and its applications to single-cell sequencing. J Comput Biol. 2012;19(5):455–77. https://doi.org/10.1089/cmb.2012.0021.
  8. Line 115. Hyperlink to raw data (NCBI Bioproject) – page not found. So, it's impossible to estimate the quality of WGS for identification of SNPs.
  9. Please, provide the description of phylogenetic tree construction (Figure 1) in Metods.
  10. Line 130. ...ST1580 (n=559). And 1580 [560] on Figure 1. Where is correct ?
  11. Line 205. two dots at the end of a sentence
  12. Lines 206 and 222. Please, indicate in italics the names of the genes
  13. Line 420. Reference #18 looks strange.

Reviewer 2 Report

In the manuscript "Alternative pathways to cirprofloxacin resistance in Neisseria gonorrhoeae..." Gonzalez et al. use serial passages to select for increased resistance to ciprofloxacin and use whole genome sequencing to monitor mutations that arise during this process. The goal is to make predictions as to what mutations will arise in natural populations. Ciprofloxacin is DNA gyrase inhibitor, and mutations arise in gyrA. The authors generate three independent lineages from each of two WHO strains. The WHO strains differ by mutations in rpoB and rpsJ and the authors are interested in why the two strains differ in their time taken to acquire resistance. A major question is whether the mutations derived from in vitrostudies are predictive of mutations that will arrive in clinical isolates. The authors point out that the strain that produces higher resistance also acquires known RAMs.

The authors use serial plating on increasing concentrations of ciprofloxacin to select for mutations conferring increased ciprofloxacin resistance. Why do they not use liquid cultures for propagation? I have to question how accurate the calculation of the number of generations can be when culturing on plates. Perhaps this experimental approach is particular to Neisseria; if so, the authors may want to point this out or otherwise justify using this technique.

The authors use whole-genome-sequencing (WGS) to monitor the appearance of mutations in gyrA and other loci throughout the genome. They find that, in addition to previously identified mutations in gyrA, the two strains acquire distinct sets of mutations, potentially explaining the higher level resistance appearing in this strain in clincial isolates. They discuss the possible role some of these mutations play in decreased sensitivity to ciprofloxacin and rationalize the appearance of these mutations.

The authors do a good job of stating the limitations of the present study, which is quite welcome and refreshing. Overall, this is an interesting study that, while leaving some questions unanswered, should be published, with some modifications.

Relatively minor issues:

One major problem is that Figure 2 is practically unreadable. It needs to be much bigger, perhaps a full page. As this is the most critical figure in the paper, it needs to be presented properly.

The data presented in the paragraph starting on Line 245 are also presented in Table S1. The authors may as well just replace this paragraph with the Table in the main text rather than as a supplemental table.

Section 3.5 would be improved if a brief concluding statement were added at the end of the section.

Line 243: This appears to be a sub-heading. It should be properly formatted as such.

Line 255: Same as comment as for Line 243.

Line 306: It is not clear what the authors mean by 'transient mutation'.
